# Effectiveness of Acupuncture in Parkinson’s Disease Symptoms—A Systematic Review

**DOI:** 10.3390/healthcare10112334

**Published:** 2022-11-21

**Authors:** Catarina Ramos Pereira, Jorge Machado, Jorge Rodrigues, Natália M. de Oliveira, Maria Begoña Criado, Henri J. Greten

**Affiliations:** 1ICBAS—School of Medicine and Biomedical Sciences, University of Porto, 4099-002 Porto, Portugal; 2CBSIn—Center of Biosciences in Integrative Health, 4000-105 Porto, Portugal; 3IPTC—Research Department in Complementary Medicine, Portuguese Institute of Taiji and Qigong, 4470-765 Maia, Portugal; 4TOXRUN—Toxicology Research Unit, University Institute of Health Sciences, CESPU, CRL, 4585-116 Gandra, Portugal; 5HSCM—Heidelberg School of Chinese Medicine, 69126 Heidelberg, Germany

**Keywords:** Parkinson’s disease, acupuncture, integrative therapy, systematic review, randomised controlled trials

## Abstract

*Background:* Parkinson’s disease (PD) is the second most common neurodegenerative disease. Several pharmacological and surgical therapies have been developed; however, they are accompanied by some adverse effects. As a result, many patients have been resorting to complementary medicine, namely acupuncture, in the hope of obtaining symptomatic improvements without having disruptive side effects. Therefore, advances in research in this area are very important. This work presents a systematic review of the effectiveness of acupuncture treatments in relieving PD symptoms. *Methods:* EMBASE, Medline, Pubmed, Science Direct, The Cochrane Library, Cochrane Central Register of Controlled Trials (Central) and Scielo databases, were systematically searched from January 2011 through July 2021. Randomised controlled trials (RCTs) published in English with all types of acupuncture treatment were included. The selection and analysis of the articles was conducted by two blinding authors through Rayyan application. *Results:* A total of 720 potentially relevant articles were identified; 52 RCTs met our inclusion criteria. After the exclusion of 35 articles, we found 17 eligible. The included RCTs reported positive effects for acupuncture plus conventional treatment compared with conventional treatment alone in the UPDRS score. *Conclusions:* Although all the studies reviewed pointed out a positive effect of acupuncture on improving motor and non-motor symptoms in Parkinson’s disease, we found great discrepancies regarding the studies’ design and methodology, making difficult any comparison between them.

## 1. Introduction

Parkinson’s disease is the second most common neurodegenerative disease worldwide, influencing both motor and non-motor symptoms. The main pathogenesis of PD is the loss of dopaminergic neurons in the substantia nigra caused by a combination of genetic and environmental factors [1,2,3]. The aetiology is still not fully understood [4]. It is a progressive pathology that affects one in every 1000 people over 60 years of age and at least 6 million people around the world [5], representing a high economic cost, causing high rates of institutionalisation and, increased health costs [6,7,8].

The pathology is characterised essentially by rest tremor, stiffness, bradykinesia, abnormal motor coordination and posture and gait changes [9,10,11]. Non-motor symptoms occur earlier and have a more profound impact on the quality of the patient’s life [8], including pain, fatigue, insomnia, anxiety, and depression [12].

Numerous pharmacological and surgical therapies have been developed to try to solve the dysfunctions derived from the pathology. However, these treatments are accompanied by some adverse effects [12]. In the clinic, conventional medical treatment consists of the internal administration of levodopa (LD), which is a symptom modifier for a limited time [3]. More than 50% of patients begin to experience LD-induced fluctuations after 2 to 5 years, beginning to develop involuntary motor movements called dyskinesias [13]. This rate increases to 80% in patients with more than 10 years of levodopa use [2]. Even though other drugs can be used, none of them is entirely efficient and is usually accompanied by a high rate of side effects [14].

Since the conventional treatment does not represent a satisfactory response, many patients have been resorting to integrative medicine, particularly acupuncture, in the hope to obtain better therapeutic outcomes without having disruptive side effects [1,13].

Acupuncture has been reported to have possible therapeutic effects in PD, as manifested by improvement in clinical symptoms such as tremors, a decrease in the dosage of antiparkinsonian drugs, a decrease in side effects, and improvements in daily life, such as improved sleep [8,15,16]. It has been found that acupuncture can protect dopaminergic neurons from degeneration via anti-oxidative stress, anti-inflammatory, and anti-apoptotic pathways as well as modulating the neurotransmitter balance in the basal ganglia circuit [3].

Prior to 2011, evidence to support acupuncture for PD symptoms remained controversial, unclear, or inconclusive due to small sample sizes, methodological failures, and poor blinding methods. Some previous systematic reviews have reported no significant effects of acupuncture due to conflicting results, whereas others have reported significant effects of acupuncture on PD symptoms [8,17]. More studies, either comparative effectiveness research or high-quality placebo-controlled clinical studies were necessary [18].

Meanwhile, randomised controlled trials (RCTs) are increasing [2,19,20], non-motor symptoms are progressively emphasised, and objective behavioural assessment tools are being employed [12]. So, it is now worth summarizing and updating the current best evidence for the use of acupuncture in PD management.

This study aimed to perform a systematic review of the literature about the safety and effectiveness of acupuncture treatments for PD symptoms based on published relevant RCTs.

## 2. Materials and Methods

### 2.1. Search Strategy and Methods for Study Identification and Screening Literature Search

The search only included articles published after 2011, to highlight recent literature and provide the most actual outcomes.

A systematic literature search was conducted from 23 June until 27 July 2021, following the PRISMA guidelines in the following databases: EMBASE, Medline, Pubmed, Science Direct, The Cochrane Library, Cochrane Central Register of Controlled Trials (Central) and Scielo.

Search terms used were based on the combination of 3 broad topics: Parkinson’s, acupuncture and RCT. Each topic included an expanded set of terms, keywords, and syntax specific to each database to achieve a wider coverage of our search. A manual search of relevant references from previous systematic reviews was also conducted. The search strategy was adjusted for each database.

### 2.2. Inclusion and Exclusion Criteria

#### 2.2.1. Types of Studies

Only clinical randomised controlled trials (RCTs) published in English were included. Were systematically searched from January 2011 through July 2021 and excluded all the articles with more than ten years.

#### 2.2.2. Participants

Patients diagnosed with PD without restrictions of age, gender, race, or duration of disease. Participants diagnosed with parkinsonian syndrome or with severe complications were excluded. Experimental studies with animal models and in vitro studies were also excluded.

#### 2.2.3. Types of Interventions

Different types of acupuncture were included: body acupuncture, scalp acupuncture, electroacupuncture or auricular acupuncture. Combined interventions with other integrative therapies that could affect the evaluation of the effectiveness of acupuncture were excluded.

#### 2.2.4. Types of Outcome Measures

The primary outcome was the efficacy rate of acupuncture on motor and non-motor PD symptoms. The efficacy rate was defined as the resolution of symptoms after treatment. The secondary outcomes were the recurrence rate and adverse events related to the treatment. All clinically relevant outcomes were eligible, and there were no restrictions for secondary outcomes.

### 2.3. Data Collection, Analysis and Management

Two reviewers (JR and NMO), separately and independently, reviewed and extracted data from each paper using a standardised data extraction form. Following the removal of duplicate items, titles and abstracts were screened to determine their eligibility based on the before-mentioned criteria, following PRISMA Guidelines. If discrepancies were encountered, a consensus would be reached by consulting an expert reviewer (BC) available for arbitration. In this case, it was not needed since a consensus was achieved for all items.

The following information was extracted: study details (authors, country, year of publication, journal, title, contact information), participants (inclusion and exclusion criteria, PD diagnostic criteria, age, gender, race, disease duration, baseline data), study methods (registry platform, sample size, blinding method, randomisation method, allocation concealment, incomplete report or selecting report), the interventions (type of acupuncture, needles, acupoints, electric frequency (if applicable), treatment duration, treatment frequency, practitioner, dosage of L-dopa) and the outcomes (primary and secondary outcomes). The corresponding authors of a particular included study were contacted for unreported or missing data.

The selection process is detailed in the Preferred Reporting Items for Systematic Reviews and Meta-Analysis flow chart shown in Figure 1.

We identified 720 publications, from Cochrane (n = 119), Science Direct (n = 461), Embase (n = 47), Scielo (n = 17) and Pubmed (n = 76); screening of the titles and abstracts reduced the number to 618, because of duplicate records.

After careful full-text screening, 563 articles were removed. The remaining 54 articles were entered into the qualitative synthesis procedure. Of these, 2 reports were not retrieved, resulting in 52 that were assessed for eligibility. We excluded 35 more studies, 29 were not RCT studies, and 6 were not in English.

Finally, 17 RCTs were included in the quantitative synthesis procedure (Figure 1). Meta-analysis was not performed as there are no comparators in the included studies.

## 3. Results

The collected data of the selected studies [15,22,23,24,25,26,27,28,29,30,31,32,33,34,35,36,37], such as participant baseline characteristics, type of acupuncture treatment, frequency of treatment, and outcome measures were narratively synthesized in Table 1.

### 3.1. Participants

#### 3.1.1. Groups

Almost all studies reviewed considered two study groups—control vs. experimental. From them, eight compared drug therapy (different kinds) with drug therapy combined with acupuncture. Similarly, one study compared *qi gong* against *qi gong* plus acupuncture treatment. Four studies used sham acupuncture vs. true acupuncture and, in another one, no intervention compared with acupuncture. One study compared healthy participants with PD patients.

To reduce placebo effects, two studies compared three groups—one study used: sham acupuncture, waiting group and true acupuncture group and another study compared: no treatment, acupuncture treatment and bee venom acupuncture treatment.

#### 3.1.2. Number of Participants

The number of participants in each study varies from 11 to 180 with a total of 845 patients in the 17 articles selected.

#### 3.1.3. Including and Excluding Criteria

Concerning including criteria we can observe a great heterogeneity concerning diagnosis type, stage of evolution of the disease, age, medication, the score of Mini-Mental State Examination, Unified Parkinson’s Disease Rating Scale (UPDRS) and fatigue scale. However, we found unanimity on conscious and communication pre-requests, signed informed consent forms and the ability to follow up.

Regarding diagnosis, seven studies based the diagnosis according to the UK Parkinson’s Disease Society Brain Bank criteria. Two studies based PD diagnosis on criteria developed by Gelb, Oliver, and Gilman [38], which is adopted by the National Institute of Neurological Disorders and Stroke, US National Institute of Health, two based the diagnosis on criteria of the Core Assessment Program for Intracerebral Transplantation (CAPIT), another one diagnosed with clinically definite idiopathic PD by a neurologist from the Kyung Hee Medical Hospital, and the last one used diagnostic criteria of the Motor Disorder and (PD) Group of the Chinese Medical Association Neurology Chapter. The other four [23,33,36], did not defined the diagnose criteria.

Concerning the stage of the disease, four studies selected I–IV stages according to the Hoehn e Yahr scale and only one selected I–III. Another study did not mention the progression of the disease according to the correspondent scale, although reported on patients that were able to walk without walking aids. The rest of the studies did not mention the progression of the disease.

In regard to the age of the participants, we found a huge divergence, however, all the studies selected adult individuals. Two studies included patients 55 years old or older, the others used wider intervals for the participant’s ages. One included from 21 to 85 years old, another from 30 to 75 years old, and another from 35 to 80 years, two opted to include ages between 40 to 75 and another from 40 to 99 years. One study only mentioned including adult participants.

Regarding medication, we found a consensus about the continuous use of anti-PD medication, however, the time of the stable dose varies from each other. Some reported at least 1 month (four articles), others 2 months (three articles), and others even 3 months (two articles). The others did not refer to the stable duration of intake of medication.

As well for the inclusion criteria, some articles mentioned the Mini-Mental State Examination with a minimum score of 18, 24 or 26 (out of 30), which reflects a state of consciousness with normal communication. Concerning the fatigue symptom, some included the presence of moderately severe fatigue as defined by a score of ≥10 on the General Fatigue Domain of the Multidimensional Fatigue Inventory or self-reported moderate or severe fatigue using the International Parkinson’s and Movement Disorder Society UPDRS fatigue item.

Additionally, two articles added inpatients or outpatients who were able to be followed up and more three signed informed consent forms.

To assess tremors at rest, all the studies used the Unified Parkinson’s Disease Rating Scale with a minimum score of more than 1 point in two or more items in the UPDRS part III.

Regarding the exclusion criteria, we found unanimity in nine articles in excluding Parkinson’s syndrome, Parkinson-plus syndrome or secondary Parkinson’s syndrome and atypical parkinsonian disease. Almost all (13 articles) excluded patients with severe previous or current psychiatric or organic neurologic disorders, patients with mental illness or dementia or any clinically psychiatric condition, cerebrovascular disease, tumour, infection, comorbidity with a bleeding disorder, severe diseases of the heart, brain, liver, kidneys, endocrine, or hematopoietic system, or even drug or alcohol abuse. Two articles excluded related gait disorders. Three articles excluded patients with Hoehn–Yahr stages 4–5.

Three articles excluded previous acupuncture therapy, one in the last 6 months. Two articles excluded non-first-consult patients. Two excluded female subjects of childbearing age or expecting a baby. One article excluded deep brain stimulation (DBS), needle phobia, comorbidity with a bleeding disorder, known anaemia with haemoglobin level < 10 g/dl, presence of symptomatic postural hypotension, age less than 45 or greater than 80 years, any contraindications for fMRI (functional magnetic resonance imaging), somatic disease, and allergic patients.

### 3.2. Randomisation

In the assessed studies, we found a great heterogeneity concerning the randomisation process. Some used non-random samples, others used the process of drawing pieces of paper from a bag, and in another, the patients were enumerated and allocated to experimental or control groups according to a simple raffle. Others used a non-blinded sample, semi-blinding (only the patients were blinded) or double-blinding method.

### 3.3. Treatment Characteristics

The treatment administered in all studies varied by type of acupuncture, acupoint selection, needle treatment technique and frequency of the treatments. This variability might have resulted in the differences that can be observed in the study’s results. We describe them below.

Nine of the seventeen studies used systemic acupuncture, four used electropuncture, four used scalp acupuncture, one auricular acupuncture and one used three techniques in the experimental protocol.

#### 3.3.1. Points

The treatment varied from 1 to 20 points. The most frequently used points (Figure 2) are GB20 (*Fengchi*), LI4 (*Hegu*), GB34 (*Yanglingquan*), used in nine, eight and seven articles, respectively. Six studies employed GV20 (*Baihui*), ST36 (*Zusanli*), LR3 (*Taichong*), SP6 (*Sanyinjiao*), five applied GV14 (*Dazhui*), and four utilised LI11 (*Quchi*).

However, we found other points not so used (Figure 3), such as GV16 (*Fengfu*), CV6 (*Qihai*), and KI3 (*Taixi*) used in three papers, LI10 (*Shousanli*), BL40 (*Weizhong*), PC6 (*Neiguan*), HT7 (*Shenmen*), Jin’s three-needle therapy, Foot motor sensory area, balance area and chorea-tremor controlled zone were applied in two different articles.

#### 3.3.2. Recurrence Rate

In some studies, the treatment frequency was once a week, for 3 weeks (3 treatments) or 8 weeks (8 treatments), but a twice a week frequency was the most commonly used during 5 weeks (10 treatments), 6 weeks (12 treatments) and three studies opted for 8 weeks (16–20 sessions) and 12 weeks (24 sessions). A frequency of 18 weeks or 36 weeks was also used. As well, a few studies opted to do the treatment 3×/week or even 4×/week for 12 weeks. Some studies just mentioned the number of treatments but not the frequency.

All studies used sterile disposable, surgical stainless steel acupuncture needles in most of the cases of 0.25 mm, however we found one study that used the 0.27 mm calibre. The length of the needles varied from 0.25 mm to 40 mm. Some authors did not reference any needle characteristics.

The needle retention time varied from 15, 20, 30 to 50 min. However, the most commonly used duration was around 30 min.

We must refer that the depth of insertion, the needle stimulation, and the needle type varied from study to study without any observable pattern, as far as we detected. In some studies, the insertion of the needle was oblique, in others transversely or even perpendicularly. The depth varies from 0.025 cm to 5 cm. The most commonly used was from 1 cm to 1.5 cm and from 2 cm to 2.5 cm in four studies. Concerning the needle stimulation, some studies stimulated and achieved the *De-Qi* (soreness, numbness, distention and pain) manually. Three studies also added the rotation movement of the needle. However, four studies added electrostimulation: one study used 4 to 100 Hz with an asymmetric biphasic square wave and pulse width of 100 microseconds (μS), another used only the 100 Hz (9 V, 1 A, 9 W) and two other studies used 2 Hz for 10 s.

#### 3.3.3. Efficacy Rate

Consensually, 16 of the 17 studies reported a positive effect of acupuncture or electropuncture on different symptoms of Parkinson’s disease. They found significant differences between the control and the experimental group and so, provided some evidence that acupuncture treatment reduces motor symptoms (reducing tremor, fatigue, improving hypsometric gait and rearranged activation of the cerebral cortex, as well as rigidity and balance), reduced non-motor symptoms (olfactory function, sleep disorders, behaviour, mood, depression and mental changes in PD), reduced complications of therapy, UPDRS scores, had a certain long-term effect and improved the quality of life.

One article [37] concluded that acupuncture may improve PD-related fatigue but looking at the results, real acupuncture offered no greater benefit than sham treatments.

### 3.4. Outcome Measure

We found a huge variation between the outcome measures used and even in the timing that they were collected.

We found that in 14 of the 17 evaluated articles, the general state of the patient was assessed utilizing the Unified Parkinson’s Disease Rating Scale (UPDRS).

However, we found a huge variation concerning other outcome measures and the timing in which they were collected. Parkinson’s Disease Quality of Life Questionnaire (PDQ) of 39 items was applied in five studies. Beck Depression Inventory (BDI), Hoehn–Yahr (H-Y) stage and Mini-Mental State Examination (MMSE) were used in four different articles. Parkinson’s Disease Sleep Scale (PDSS) was applied in three different studies. The SF-12 health survey was applied in two different articles, as well as the WHO quality of life (WHOQOL), Short Falls Efficacy Scale-International (FES-I), Visual Analogue Scale (VAS), Modified Webster Scale, Self-rating Depression Scale (SDS), Nonmotor Symptoms Quest (NMSQ), Hamilton Depression Scale (HAMD), Pittsburgh Sleep Quality Index (PSQI), Hamilton Anxiety Scale (HAMA), General Fatigue Score of the Multidimensional Fatigue Inventory (MFI-GF), fMRI scans of the patient’s brains. Geriatrics Depression Scale (GDS), Epworth Sleepiness Scale (ESS), MRI test results, Edinburg Handedness Inventory, Montreal Cognitive Assessment, Berg Balance Scale, Test of Smell Identification (TSI), Gait Disturbance (PIGD) Score, gait speed and number, postural stability, neurotransmitter levels, Modified Fatigue Impact Scale, Epworth Sleepiness Scale (ESS), Apathy Evaluation Scale (AES) were used in just one article as well as steady-state gait speed (stride length, cadence, double support, and midswim angular velocity), neuroinflammatory factors: nitric oxide, tumour necrosis factor, interleukin-1, and prostaglandin, neurotransmitters: dopamine, acetylcholine, norepinephrine, and 5-hydroxytryptamine, balance assessment (medial-lateral centre of gravity sway to anterior-posterior sway and ankle-to-hip sway during eyes-open, eyes-closed, and eyes-open dual-tasks trials), time and number of steps required to walk 30 m, posturography (Balance Master System) and even gait parameters—GAITRite system and hemodynamic responses in the cerebral cortices using functional near-infrared spectroscopy.

The fact that different articles used different scales makes it very difficult to compare the results obtained.

### 3.5. Adverse Events

Of the 17 articles, 3 reported side effects, 10 reported no side effects and 4 did not refer to the presence or absence of adverse events. The side effects described were: one patient in the experimental group reported transient light-headedness at the end of the final treatment, one patient described constipation and the other reported a total of three adverse events not related to the acupuncture treatment.

## 4. Discussion

Previous systematic reviews about the effectiveness of acupuncture in Parkinson’s disease included studies conducted before 2011. Most of these studies had several limitations, namely regarding the study design and methodology used, which made it difficult to draw definitive conclusions [2,39,40]. However, although the evidence had been inconclusive, the therapeutic potential of acupuncture in Parkinson’s disease seemed to be quite promising [41,42]. Thus, the present systematic review including renewed literature with recent RCTs, a large sample size and better-quality studies would be important to provide better evidence on the efficiency of acupuncture in Parkinson’s disease symptoms [43].

According to the studies analysed in the present systematic review, which includes studies published between 2011 and 2021, there seems to be evidence for a positive effect of acupuncture on improving motor and non-motor symptoms in Parkinson’s disease. In all the studies reviewed, acupuncture was more effective in alleviating the symptoms of Parkinson’s disease than no treatment or conventional pharmacological treatment alone [43]. Additionally, some studies found that when acupuncture was used as an adjuvant to pharmacology, namely levodopa, it improved its therapeutic efficacy, allowing a reduction in the dose and the occurrence of adverse effects arising from its use [40]. There seem to be no differences in the effectiveness concerning the type of acupuncture used (body acupuncture, electroacupuncture or scalp acupuncture) but we did not find studies on the comparison of different Chinese medicine approaches [39].

Although all the studies reviewed pointed out a positive effect of acupuncture, we found great discrepancies regarding the studies’ design and methodology [2,42]. This lack of standardisation made it difficult to compare them [39,40,43].

Most studies reviewed included two study groups—comparing the control group (conventional pharmacological treatment) with the experimental group (acupuncture per se plus conventional treatment). In these studies, the issue of placebo raises, since the patients realised which individuals are included in the experimental group and the control group, and what can affect the patient’s final response through the placebo effect. This is one of the most present issues in Chinese medicine research, and the most difficult to circumvent and minimise since it makes it difficult to prove that the observed improvement refers to the effect of acupuncture *per se* and not to the placebo effect. To address this issue, one revised study distributed the sample into three groups obtaining evidence that the positive effects of acupuncture were not due to the placebo effect. In addition, the study [27] using functional magnetic resonance imaging (fMRI) as an objective measure, showed that acupuncture reduces tremors in patients with Parkinson’s disease.

The fact that some studies were not double-blind and only presented a semi-blinding or a no-blinding methodology greatly affects the ability to validate the results obtained, which is another major constraint.

We also found numerous methodological flaws [2,39], particularly in terms of the selection of points. Most of the studies included in this systematic review assessed the effectiveness of a group of points, which were chosen based on the theoretical support of traditional Chinese medicine. However, the group of points was different in all studies, as well as the respective theoretical support, which limits the comparison between them. This makes it equally difficult to provide evidence on which point contributes to the improvement of which symptom. Although traditional Chinese medicine professionals document their clinical practice, there is still little literature involving the parameters of the selection of meridians and standardised acupoints [9]. In this systematic review, only one study evaluated the effectiveness of a single point [28].

In addition to the selection of points, differences were also found in the frequency of treatments, the number of treatments, the time the needle remains in the body, the type of needle, the stimulation of the needle and the depth at which it was inserted [44].

This raises yet another question. Most studies did not assess a particular symptom, but a general health status, through qualitative methodologies. The only instrument widely used at a methodological level was the UPDRS scale. This fact, once again, is a limiting factor to obtain evidence on which symptoms can be improved by acupuncture.

Important aspects that are not yet clarified, involve, on one hand, the correlation between the improvement in performance between motor and non-motor symptoms, which usually precede them. Studies on the non-motor effects of Parkinson’s disease, such as olfactory changes, sleep disorders, behavioural or mood changes and depression were developed, however, few considered the influence that these had on motor symptoms. On the other hand, there are still no studies on the correlation between the improvement of symptoms and the respective changes at the cortical level, assessed by functional magnetic resonance imaging.

Regarding the mechanisms of action of acupuncture, some studies indicated that acupuncture has neurotrophic and neuroprotective effects [10], others, using functional magnetic resonance, indicated that acupuncture activates the putamen and the primary motor cortex, and these activations were correlated with the most appropriate motor function [45], suggesting additional involvement of the posterior medial cortex and temporal cortex in the central effect of acupuncture, called “limbic-paralimbic-neocortical network” [13]. Recently, Zhao et al. found that, particularly during the early stages, acupuncture may reduce the neurodegeneration of dopaminergic neurons and regulates the balance of the dopaminergic circuit, thus delaying the progression of the disease [3].

In our opinion, further studies should be carried out to assess and explore these aspects to provide consistency and decode the basis of motor and non-motor alterations. It should be noted that few studies have addressed the long-term effects of acupuncture treatments, which will give medical consistency to the application of this therapy in this population spectrum. Future studies should address this issue, to understand how long the improvements can be maintained when over time.

Thus, considering the limitations of what had been published until now, we propose that a standardised intervention program should be created, with support for the selection of the points to be used to minimise the progression of Parkinson’s disease and its respective symptoms. Furthermore, it would also be pertinent to standardise how the outcomes are collected, bearing in mind that the computerised instruments to collect the data must be portable, but valid and reliable, collecting data quantitatively and not merely qualitatively, as is the case in most studies published. In that way, they could be applied not only in research but also in clinical practice. Thus, it would be able to build more studies that could positively help the evolution of existing knowledge in this area.

It would also be pertinent to carry out a comparative study of the various approaches of Chinese medicine, to try to understand which would present the highest percentage of improvement—scalp acupuncture, electropuncture or body acupuncture. Thus, it would also contribute to minimising discrepancies in pre-existing treatments.

It is also our opinion that studies with larger sample sizes and double-blind methodologies are needed [43]. Finally, it would be also very important to promote the performance of studies with higher quality, following STRICTA and/or CONSORT recommendations, since most of the existent bibliography lacks it [2,43].

## 5. Conclusions

In conclusion, although recent research provides evidence for the positive effects of acupuncture on PD symptoms, we think that further studies addressing the following aspects are essential. First, the safety of acupuncture should be evaluated and reported [18]. Second, the long-term effects must be measured. Third, objective assessments using portable novel computerised technologies should be considered. Fourth, comparative studies should be carried out about the differences in effectiveness among scalp acupuncture, electropuncture and systemic acupuncture, to see which one is more effective in reducing PD symptoms. Fifth, the correlation between the changes in the symptoms and neurological changes should be investigated. Sixth, target symptoms should be selected and evaluated instead of only performing global evaluations. Seventh, large, multicentre, well-designed RCTs should be organised for evaluation of the efficacy of acupuncture.

## Figures and Tables

**Figure 1 healthcare-10-02334-f001:**
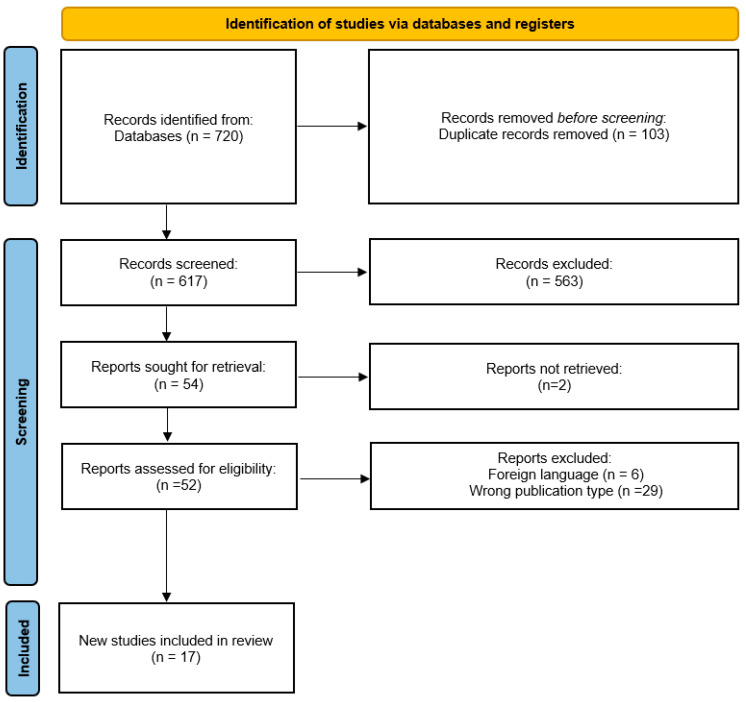
Flow diagram of the trial selection process. Extracted and adjusted from PRISMA [21].

**Figure 2 healthcare-10-02334-f002:**
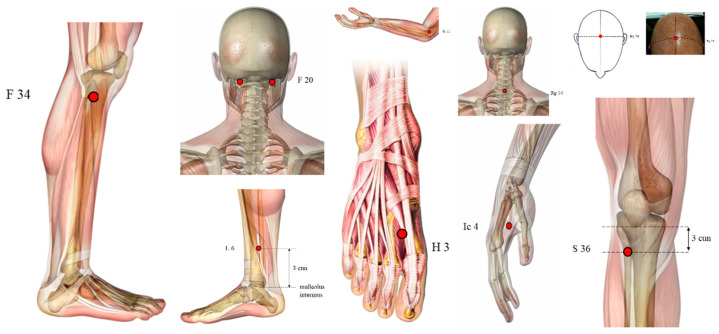
Ilustration of the most used points (highlighted in red dots): GB20 (*Fengchi*), LI4 (*Hegu*), GB34 (*Yanglingquan*), GV20 (*Baihui*), ST36 (*Zusanli*), LR3 (*Taichong*), SP6 (*Sanyinjiao*), GV14 (*Dazhui*), and LI11 (*Quchi*).

**Figure 3 healthcare-10-02334-f003:**
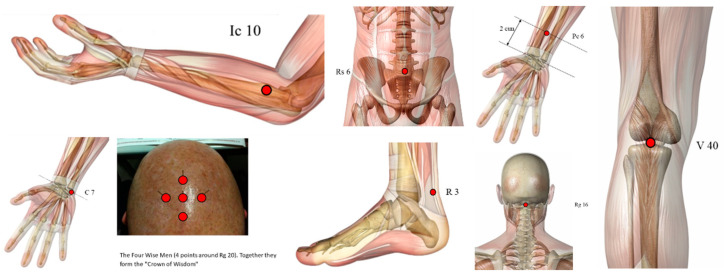
Illustration of other points used (highlighted in red dots): GV16 (*Fengfu*), CV6 (*Qihai*), KI3 (*Taixi*), LI10 (*Shousanli*), BL40 (*Weizhong*), PC6 (*Neiguan*), HT7 (*Shenmen*). Some acupoints (Figure 4) such as TE5 (*Waiguan*), EX-HN-1 (*Sishencong*), BL23 (*Shenshu*), BL18 (*Ganshu*), ST32 (*Futu*), GB38 (*Yangfu*), GB31 (*Fengshi*) and GV24 (*Shenting*) were used only by one study.

**Figure 4 healthcare-10-02334-f004:**
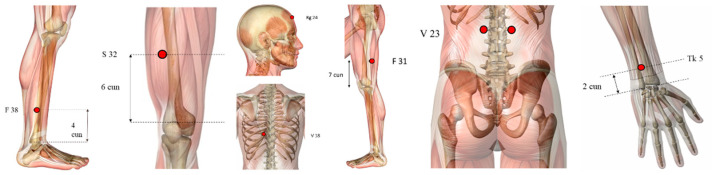
Illustration of the less frequent points used (highlighted in red dots): TE5 (*Waiguan*), EX-HN-1 (*Sishencong*), BL23 (*Shenshu*), BL18 (*Ganshu*), ST32 (*Futu*), GB38 (*Yangfu*), GB31 (*Fengshi*) and GV24 (*Shenting*).

**Table 1 healthcare-10-02334-t001:** Collected data (Legends: NA = not applicable; EA = electroacupuncture; PD = Parkinson’s disease; fMRI = functional magnetic resonance imaging).

Study Details	Participants	Study Design	Control Group	Interventions (GE)	Outcomes Measures	Results/Conclusions	Side Effect
A Clinical Study of Integrating Acupuncture and Western Medicine in Treating Patients with Parkinson’s Disease *The American Journal of Chinese Medicine* [22]	**Inclusion Criteria:** Diagnosed with PD for over 6 months (criteria of the Core Assessment Program for Intracerebral Transplantation (CAPIT). **Exclusion Criteria:** Severe diseases of the heart, brain, liver, kidneys, endocrine, or hematopoietic system, psychosis, or dementia and who did not agree to participate in the study. **Dropout:** NA **Randomisation:** Non-random sample	Effects of acupuncture treatment in PD patients N = 40	Drug Therapy Levodopa N = 20	Acupuncture Treatment + Drug Therapy **19 points:** DU20 (unilateral), GB20, LI11, LI10, LI4, GB31, ST32, GB34, GB38 (bilateral) Frequency: 2×/week - short-term (18 weeks—36 sessions) n = 20 - long-term (36 weeks—72 sessions) n = 13 Depth of insertion: 5–30 mm Needle stimulation: manual (“De-Qi”: soreness, numbness, distention and pain) Time: 15 min Needle type: 0.27 mm × 25/40 mm	**3 Assessment Moments:**before, in the middle and at the end of the study. - Unified Parkinson’s Disease Rating Scale (UPDRS) - Beck Anxiety Inventory (BAI) - Beck Depression Inventory-Version 2 (BDI-II) - WHO quality of life (WHOQOL)	Acupuncture treatment: -reduces symptoms and signs of mind, behaviour, mood, depression - complications of therapy - reduce UPDRS scores - improve the quality of life	NO
A Pilot Clinical Trial to Objectively Assess the Efficacy of Electroacupuncture on Gait in Patients with Parkinson’s Disease Using Body Worn Sensors *PLoS ONE* [23]	**Inclusion Criteria:** (1) Community-dwelling aged 55 years or older with a diagnosis of PD; (2) patients who can walk 20 m without walking assistance; and (3) patients who are stable without anti-PD medication(s) change for at least 1 month. **Exclusion Criteria:** Patients who (1) have received previous acupuncture; (2) DBS; (3) with any clinically significant medical condition, psychiatric condition, drug or alcohol abuse, or laboratory abnormality that would, in the judgment of the investigators, interfere with the ability to participate in the study; and (4) with non-PD related gait disorders **Dropout:** zero dropout rate **Randomisation:** drawing pieces of paper from a bag	Efficacy of EA for gait disorders using body-worn sensor technology in patients with PD N = 15	Sham Acupuncture Insertion less than 4 mm, under the skin at non-acupuncture points, without needle manipulation N = 5	EA Treatment **11 points:** in Foot Motor Sensory Area, Balance Area, GV20, GV14, LI4, ST36, GB34, BL40, SP6, KI3, LR3 Frequency: 3×/week Depth of insertion: 0.25 × 40–50 mm—Transversely in GV20, Balance Area, Foot Motor Sensory Area and Perpendicularly in the rest Needle stimulation: from 4 to 100 Hz. The pulse was an asymmetric biphasic square wave with a pulse width of 100 microseconds (μS). “De-qi” was achieved with needle manipulation. Three pocket portable electric stimulators (ITO ES-130, Japan) were used for EA stimulation. Time: 30-min sessions Needle type: Sterile disposable, surgical stainless steel acupuncture needles (Seirin, L type, Japan) N = 10	**2 Assessment Moments:** At baseline and after completion of treatments. - Unified Parkinson’s Disease Rating Scale (UPDRS) - SF-12 health survey - Short Falls Efficacy Scale-International (FES-I) - Visual analogue scale (VAS) - Steady-state gait speed (stride length, cadence, double support, and midswing angular velocity)	All gait parameters were improved in the experimental group in response to EA treatment.	YES
Acupuncture as Adjuvant Therapy for Sleep Disorders in Parkinson’s Disease *Journal of Acupuncture and Meridian Studies* [15]	**Inclusion criteria:** (1) patients with idiopathic PD according to the UK Parkinson’s Disease Society Brain Bank criteria (2) Stage I e III PD, according to the Hoehn e Yahr scale (3) age 35 e 80 years; (4) minimum score of 18 on the Mini-Mental State Examination for a low academic level or a minimum score of 26 for high academic level and (5) a stable dose of antiparkinsonian medication for 2 months. **Exclusion criteria:** another neurological condition or who had received physiotherapy **Dropout:** NA **Randomisation:** The patients were enumerated and allocated to experimental or control groups according to a simple raffle.	Effects of acupuncture on sleep disturbances N = 11	No Intervention **N = 11**	Acupuncture Treatment **8 points:** LR3, SP6, LI4, TE5, HT7, PC6, LI11, GB20. Frequency: 1×/week for 8 weeks Time: 30 min N = 11	**2 Assessment Moments:**Prior to the start of the study and after. - MMSE - HY scale - PDSS (Parkinson’s Disease Sleep Scale)	A therapeutic benefit of acupuncture in sleep disorders in patients with PD.	NO
Madopar combined with acupuncture improves motor and non-motor symptoms in Parkinson’s disease patients: A multicentre randomised controlled trial. *European Journal of Integrative Medicine* [24]	**Inclusion:** (1) met the diagnostic criteria; (2) aged between 40–75 years; (3) patients with the revised Hoehn–Yahr stage 1–4; (4) Madopar dosage could be maintained for ≥3 months with a stable medical condition or first-consult patients who had not been treated for Parkinson’s disease in the past; (5) inpatients or outpatients who were able to follow up; (6) conscious with normal communication; and (7) signed informed consent form. **Exclusion:** Patients with (1) Hoehn–Yahr stage 4–5; (2) Parkinson’s syndrome (3) merged with other serious diseases, such as diseases of the heart, liver, kidney, endocrine and hematopoietic system; (4) mental illness or dementia; (6) allergic patients; or (7) non-first-consult patients.	Effect of integrated therapy on improving motor and nonmotor symptoms in PD patients. N = 76	Drug Therapy Madopar only	Acupuncture Treatment + Drug Therapy (Madopar) Jin’s three-needle therapy: GV17, GB19^A^, Sishenzhen and temporal three-needle midpoint of the posterior hairline, the depression above the *protuberantia occipitalis externa*. Frequency: 4 days per week; 12 weeks—2×/week (four-week follow-up) Time: 30 min per day	**3 Assessment Moments:** In Week 0. within (Week 4), and after (Week 8) - Unified Parkinson’s Disease Rating Scale (UPDRS) - Modified Webster Scale - Parkinson’s Disease Sleep Scale (PDSS) - Self-rating Depression Scale (SDS)	Integrated therapy showed a greater advantage, with earlier non-motor symptom improvement. Acupuncture had a certain long-term effect.	NO
Acupuncture for motor symptom improvement in Parkinson’s disease and the potential identification of responders to acupuncture treatment *Integrative Medicine Research* [25]	**Inclusion:** a stable dosage of anti-PD medication for at least two months without adverse effects and diagnosed by a neurologist according to the UK Parkinson’s Disease Society Brain Bank criteria **Exclusion:** Parkinson-plus syndrome or secondary Parkinson’s syndrome	Effectiveness of EA and explore its mechanisms in PD patients when used as adjunctive therapy to conventional drugs. N = 50	Drug Therapy	EA + drug **4 points:** GB20 + LI4 and central GV16 + GV14 Frequency: 8 weeks—20 session sessions that occur every 3 days Needle stimulation: 9 V, 1 A, 9 W, and 100 Hz Time: 30 min	**Assessment Moments:** 12 h after the latest intake of medication. - Unified Parkinson’s Disease Rating Scale (UPDRS) III and IV - Hoehn–Yahr (H-Y) stage -Nonmotor Symptoms Quest (NMSQ) -Parkinson’s Disease Quality of Life Questionnaire (PDQ) 39 items - Hamilton Depression Scale (HAMD) - Pittsburgh Sleep Quality Index (PSQI) - Hamilton Anxiety Scale (HAMA) - Neuroinflammatory factors: nitric oxide, tumour necrosis factor, interleukin-1, and prostaglandin –Neurotransmitters: dopamine, acetylcholine, norepinephrine, and 5-hydroxytryptamine.	Acupuncture improves motor symptoms and quality of sleep	NO
Acupuncture in the treatment of fatigue in Parkinson’s disease: A pilot, randomised, controlled, study. *Brain and Behaviour* [26]	**Inclusion:** (1) diagnosis of PD based on criteria developed by Gelb, Oliver, and Gilman (1999) which is adopted by the National Institute of Neurological Disorders and Stroke, US National Institute of Health, (2) age 21–85 years old, (3) presence of moderately severe fatigue as defined by a score of ≥10 on the General Fatigue Domain of the Multidimensional Fatigue Inventory (Smets, Grasen, Bonke, and De Haes, 1995), and (4) no acupuncture treatment in the past 6 months. **Exclusion:** (1) significant cognitive, language or psychiatric illnesses which prevent the subject from understanding instructions and participating in the study, (2) needle phobia, (3) comorbidity with a bleeding disorder, (4) known anaemia with haemoglobin level < 10 g/dl, (5) known congestive cardiac failure and/or end-stage renal disease, (6) female subjects of childbearing age, and (7) presence of symptomatic postural hypotension.	Efficacy of acupuncture in patients with PD-related fatigue N = 40	Sham Acupuncture Needles developed by Jongbae Park (Park Sham Device, PSD)—When pressed onto the skin, it telescopes into the handle and the blunt tip stay on the skin instead of penetrating it. The plastic tube with adhesive foot-plate is placed on the skin to hold it in place	Real Acupuncture **11 acupoints:** right PC 6, left PC 6, right LI 4, left LI4, right ST 36, left ST 36, right SP 6, left SP 6, right KI 3, left KI 3, and CV 6. Frequency: 10 sessions—5 weeks—2×/week Time: 20 min	**3 Assessment Moments:** at baseline, 5 and 9 weeks. - General Fatigue Score of the Multidimensional Fatigue Inventory (MFI-GF) - MFI-Total Score - Unified Parkinson’s Disease Rating Scale Motor score (UPDRS Motor) - Parkinson’s Disease Questionnaire-39 (PDQ 39) - Geriatrics Depression Scale (GDS) - Epworth Sleepiness Scale ESS)	Both real and sham acupuncture are equally effective in improving PD-related fatigue, and this is likely due to nonspecific or placebo effects this improvement was maintained up to 4 weeks after completion of treatment.	YES—3
Acupuncture Modulates the Cerebello-Thalamo-Cortical Circuit and Cognitive Brain Regions in Patients of Parkinson’s Disease with Tremor *Frontiers Aging Neuroscience* [27]	**Inclusion:** diagnosed based on the UK PD Society Brain Bank clinical diagnostic criteria, and tremor at rest in at least one upper or lower extremity on either side was assessed by item 20 of the Unified Parkinson’s Disease Rating Scale **Exclusion:** secondary Parkinsonism, atypical parkinsonian disease, advanced PD stage (H-Y ≥ 4), age less than 45 or greater than 80 years, history of other neurological disorders or head trauma, left-handedness, cognitive impairment (Mini-Mental State Examination (MMSE) score < 24), depression tendency (Beck Depression Inventory (BDI) score > 4), and any contraindications for fMRI.	Effect of acupuncture on PD patients with tremor and its potential neural mechanism by fMRI N = 41	Sham Acupuncture (0.2 cm deep and 0.5cun next to the points) + Waiting Group (True acupuncture was performed after final evaluation)	True acupuncture group DU20, GB20, and the Chorea-Tremor Controlled Zone Needle stimulation: The reinforcing-reducing method conducted by twirling was performed every 10 min Time: 30 min needle retention time	**2 Assessment Moments:**Before and after the treatment course. - fMRI scans of the patients’ brains - UPDRS II and III subscales	Acupuncture reduces tremor	NO
Acupuncture on GB34 activates the precentral gyrus and prefrontal cortex in Parkinson’s disease *BMC Complementary and Alternative Medicine* [28]	**Inclusion:** Diagnosed with clinically definite idiopathic PD by a neurologist from the Kyung Hee Medical Hospital. **Exclusion:** Atypical parkinsonian disorder, other neurological or major medical conditions (head injury, stroke) or current psychiatric problems.	Evaluate the influence of GB 34 in brain areas And compare brain activity between PD patients and healthy participants N = 24	Healthy Participants N = 12	PD Patients **1 Point:** Right GB 34 (frequently used acupoint for motor function treatment in the oriental medical field needle) Depth of insertion: 1.0 cm Needle type: 0.25 × 40 mm N = 12	**Assessment Moments** - fMRI experiment - Edinburg Handedness Inventory - Hoen and Yahr stage - Korean Mini-Mental State Examination - Beck Depression Inventory	GB 34 seems to be a suitable acupoint	NO
Does Integrative Medicine Enhance Balance in Aging Adults?—Proof of Concept for Benefit of Electro-acupuncture Therapy in Parkinson’s Disease *Gerontology* [29]	**Inclusion:** Aging adults aged 55 years or older with idiopathic PD diagnosed by movement disorder specialists based on the UK Brain Bank criteria. **Exclusion:** Any type of neurological disorder other than PD or if they have prior experience with EA therapy.	Assess EA improvement in postural balance in PD patients N = 56	Sham Acupuncture (With the insertion of needles just under the skin at non-acupuncture points) stimulation with minimal intensity compared to real EA (just turning on the light of the stimulator) Healthy Participants	**EA treatment** **20 points:** GV20, GV14 on the midline and bilateral Foot Motor Sensory Area, Balance Area, bilateral ST36, LI4, GB34, LR3, KI3, SP6, BL40. Frequency: 1×/week—3 weeks Needle stimulation: Electrical stimulation frequency: 4 Hz or 100 Hz with intensity just below the level that induces visible muscle contraction. Time: 30 min N = 15	**2 Assessment Moments:** At baseline and after the final therapy. In the “off medication stage” (>12 h after their last PD medication dose). - Balance assessment (Medial-lateral centre of gravity sway to anterior-posterior sway and ankle-to-hip sway during eyes-open, eyes-closed, and eyes-open dual-tasks trials) - Unified Parkinson’s Disease Rating Scale (UPDRS) - Visual analogue scale (VAS) - SF-12 health survey - Short Falls Efficacy Scale-International (SHORT FES-I) - Mini-Mental State Examination (MMSE) - Hoehn and Yahr staging	Improvement in rigidity and balance following EA	NO
Effect and Potential Mechanism of Electroacupuncture Add-On Treatment in Patients with Parkinson’s Disease *Evidence-Based Complementary and Alternative Medicine* [30]	**Inclusion:** a stable dose of anti-Parkinsonian medication for at least 2 months and did not report adverse events. Patients were diagnosed with PD according to UK Parkinson’s Disease Society Brain Bank criteria. **Exclusion:** secondary Parkinson’s syndromes, Parkinson-plus syndromes, infectious disease in central and peripheral systems, dysarthria, severe psychiatric diseases affecting expression, malignant tumour, disability, and other serious somatic diseases.	Effectiveness and mechanisms of EA add-on treatment in PD patients N = 50	Drug Therapy	Drug + EA **6 points:** bilateral GB20 and LI4 and central Du14 and Du16 Frequency: 20 sessions every 3 days and lasted 2 months Depth of insertion: 2.0–2.5 cm Needle stimulation: inserted obliquely electrical pulses of 9 V, 1 A, 9 W, and 100 Hz Time: 30 min Needle type: 0.25 mm × 40 mm	**2 Assessment Moments** Before and after. GC was reevaluated on the first day after 2 months of drug treatment. Evaluated 12 h after Parkinsonian drugs. - Unified Parkinson’s Disease Rating Scale (UPDRS) III - Hamilton Depression Rating Scale (HAMD-) - Pittsburgh Sleep Quality Index - Anxiety Scale- (HAMA-) - Nonmotor Symptoms Quest (NMSQ) - Montreal Cognitive Assessment - Mini-Mental State Examination	EA dramatically reduces motor symptoms of PD patients and nonmotor symptoms, especially sleep quality and depression	NO
Effectiveness of acupuncture and bee venom acupuncture in idiopathic Parkinson’s disease *Parkinsonism and Related Disorders* [31]	**Inclusion:** adults with idiopathic Parkinson’s disease who had been on a stable dose of antiparkinsonian medication for at least 1 month. The study neurologist diagnosed each patient with IPD based on symptoms, medications, and brain imaging. **Exclusion:** patients with severe previous or current psychiatric or organic brain disorders other than PD (including secondary Parkinsonism), atypical Parkinsonism, somatic diseases, alcohol abuse, or narcotic abuse.	Effectiveness of both acupuncture and bee venom acupuncture as adjuvant therapies for idiopathic Parkinson’s disease N = 35	No Intervention N = 9	1. Acupuncture Treatment 2. Bee Venom Acupuncture Treatment **10 points:** GB 20, LI11, GB34, ST36, LR3 bilateral Frequency: 16 total sessions—twice a week for 8 weeks Depth of insertion: 1.0 × 10^1.5^ cm Needle stimulation: rotated at 2 Hz for 10 s to achieve Deqi Time: 20 min. Needle type: 0.25mm × 0.30 mm N = 13 each	**2 Assessment Moments** Before and after the treatment. - Unified Parkinson’s Disease Rating Scale - Parkinson’s Disease Quality of Life Questionnaire - Beck Depression Inventory - Berg Balance Scale - Time and number of steps required to walk 30 m	Both acupuncture and bee venom acupuncture showed promising results as adjuvant therapies for Parkinson’s disease	NO
Effects of Acupuncture and Qigong Meditation on Nonmotor Symptoms of Parkinson’s Disease *Journal of Acupuncture Research* [32]	**Inclusion:** idiopathic PD (based on UK Parkinson’s Disease Society Brain Bank criteria), who were taking an anti- Parkinson’s medication **Exclusion:** NA	Whether Qigong and acupuncture affect nonmotor symptoms of PD N = 21	Qigong Meditation Only	Acupuncture and Qigong Meditation **6 points:** bilateral GB20, LI4 and central Du14 and Du16 + Healing breathing, Kwanjeong Meditation, and Qigong healing—repeated 12 times Frequency: 12 treatments—5 min until De qi was achieved Depth of insertion: 2.0–2.5 cm obliquely Time: 50 min Needle type: 0.25 × 40 mm	**2 Assessment Moments** Before and after 12 treatments at baseline and 1 month after 12 treatments. - Unified Parkinson’s Disease Rating Scales (UPDRS 1) - Test of Smell Identification (TSI)	The combination of Qigong and acupuncture and Qigong alone was shown to improve the nonmotor symptoms and olfactory function of PD.	NA
Efficacy of Combined Treatment with Acupuncture and Bee Venom Acupuncture as an Adjunctive Treatment for Parkinson’s Disease *The Journal of Alternative and Complementary Medicine* [33]	**Inclusion:** Stable dose of antiparkinsonian medication for at least 1 month; Hoehn and Yahr scale I–IV; More than 1 point in two or more items in the UPDRS part III; Mini-Mental State Examination-Korean version score greater than 24 points (out of 30); Consent to the study after full description. **Exclusion:** Severe psychiatric or organic brain disorders other than PD, previous or current; Secondary parkinsonism due to medication, cerebrovascular disease, tumour, infection, etc; Atypical Parkinsonism or Parkinson-plus syndrome; Somatic disease; Alcohol or narcotic abuse; Expecting a baby; Determined to be inappropriate for participation by the investigator. **Dropout:** Skipped more than 8 of the total 24 treatment sessions in the study or control group; Not able to continue the study because of serious adverse events or aggravation of the condition; Withdrawal of the consent; Impossible to complete the scheduled study as planned by the judgment of the principal investigator. **Randomisation:** Double-blind	Efficacy of acupuncture and bee venom acupuncture for idiopathic Parkinson’s disease. Duration of the effects by follow-up assessments at the end of the treatment N = 63	Drug Therapy Antiparkinsonian drugs without additional intervention Sham Acupuncture received a normal saline injection (shallow minimal acupuncture stimulation was performed at the same points using the same acupuncture needles for 15 min without de qi)	Acupuncture Treatment **10 points:** GB20, LI11, GB34, ST36, and LR3 bilateral Frequency: twice a week for 12 weeks Depth of insertion: 1.0–1.5 cm Needle stimulation: rotated at 2 Hz for 10 s to achieve de qi Time: 15 min Needle type: 0.25 × 30 mm	**4 Assessment Moments** At baseline and at 12, 16, and 20 weeks. Performed in the medication “on” - Unified Parkinson’s Disease Rating Scale (UPDRS) part II and part III score - Postural instability—posturography using the Balance Master System - Gait disturbance (PIGD) score - Gait speed and number - Parkinson’s Disease Quality of Life Questionnaire - Beck Depression Inventory	Combined treatment of acupuncture and BVA might be a safe and useful adjunctive treatment	NO
Comparative study of the efficacy of the usual therapy for Parkinson’s disease plus auricular acupuncture and the usual therapy without acupuncture *Revista Internacional de Acupuntura* [34]	**Inclusion:** Diagnosis of idiopathic PE confirmed by a neurologist. Symptomatic PE with a duration of between 1 and 15 years before entering the study. Age between 30 and 75 years at the time of diagnosis. **Exclusion:** Major motor disabilities (only a la EP). Psychiatric illnesses such as: Clinically diagnosed depression (severe depression will be considered to have 30 points or more on the Beck scale), generalised anxiety, social phobia, etc.—Cortical dementia with synamnesia, aphasia, or any other typical Alzheimer’s symptom.—Parkinsonism (parkinsonian syndromes), isolated phenomena, cerebellar syndrome.	Efficacy of the routine treatment in Parkinson’s disease (PD) plus permanent auricular acupuncture versus routine treatment without acupuncture N = 32	Sham Acupuncture	100–120 needles Depth of insertion: 2 mm Needle type: titanium needles	**2 Assessment Moments** Before and 12 weeks after. - Unified Parkinson’s Disease Rating Scale	The efficacy of the *Forever Needles*, compared to those of the group without permanent insertion for the treatment of PD, could not be established in this study.	NA
Evaluation of Rehabilitation and MRI Results of the Combined Therapy of *Bushenzhichan* Formula and Needle Embedding for Parkinson’s Disease *Indian Journal of Pharmaceutical Sciences* [35]	**Inclusion:** Diagnostic criteria of the Motor Disorder and (PD) Group of the Chinese Medical Association Neurology Chapter **Exclusion:** NA	Evaluate Parkinson’s disease patients using combined therapy of *Bushenzhichan* formula and needle embedding and analysing the Magnetic resonance imaging test results N = 180	Drug Therapy Madopar Only (100/1500 mg, 3–4 times a day) N = 90	Madopar as well as combined therapy of *Bushenzhichan* formula and needle embedding. The constituents of *Bushenzhichan* formula are as follow up 9 g deer antler glue, 9 g *Zhigancao*, 20 g raw tortoise plastron, 20 g *fructus cannabis*, 20 g *codonopsis*, 20 g raw turtle shell, 12 g gelatin, 12 g *gastrodiaelata*, 12 g pulp of cornus, 30 g *Fructus Lycii*, 30 g *Radix Paeoniae alba*, 30 g dries *Radix Rehmanniae*, 30 g *cistanche*, 30 g *uncaria*, 15 g *Radix Ophiopogonis*, 10 g *polygonatum*. It is considered a single Chinese herb free from decoction, which could be dissolved in 400 mL of boiled water, once a day, and used as a tea after lunch and dinner while warm **Points:** *Baihui (GV 20)*, *Sishencong (EX-HN 1)*, *Fengchi (GV 20)*, *Taichong (LR 3)*, *Yanglingquan (GB 34)*, *Sanyinjiao (BP6)*, *Hegu (LI4)*, *Ganshu (BL18)*, *Shenshu (BL23)* and the chorea-trembling controlled area. N = 90	**Assessment Moments** - MRI test results - UPDRS	The combined therapy of *Bushenzhichan* formula and needle embedding could significantly improve the therapeutic efficacy and promote recovery in Parkinson’s disease patients	NA
Gait Disturbance Improvement and Cerebral Cortex Rearrangement by Acupuncture in Parkinson’s Disease: A Pilot Assessor-Blinded, Randomised, Controlled, Parallel-Group Trial *Neurorehabilitation and Neural Repair* [36]	**Inclusion: WI** **Exclusion: WI** **Randomisation:** Not blinding	Effects of acupuncture on gait disturbance N = 26	Drug Therapy	Acupuncture Treatment + Drug Therapy 8 treatments—2×/week for 4 weeks **Points:** WI	**Assessment Moments** Assessments in the “on” state intervention phase (4 weeks) and a follow-up phase (4 weeks). - Gait parameters—GAITRite system and hemodynamic responses in the cerebral cortices using functional near-infrared spectroscopy - Unified Parkinson’s Disease Rating Scale (UPDRS) scores - Neurotransmitter levels	Acupuncture tended to improve hypometric gait and rearranged activation of the cerebral cortex	NA
Randomised, Controlled Trial of Acupuncture for Fatigue in Parkinson’s Disease *Movements Disorders* [37]	**Inclusion:** UK Brain Bank criteria for probable PD10; age 40 to 99 years; fluent in English; stable medication uses for 30 days; and self-reported moderate or severe fatigue using the International Parkinson’s and Movement Disorder Society UPDRS fatigue item. **Exclusion:** dementia or a score below 24 on the Montreal Cognitive Assessment (MoCA) presence of another medical condition expected to produce fatigue; active depression or Hospital Anxiety and Depression Scale (HADS) depression subscale score greater than 10; the presence of an untreated sleep disorder; or exposure to acupuncture within the past 6 months.	Determine whether a 6-week course of acupuncture could improve PD-related fatigue N = 94	Sham Acupuncture (non-acupuncture points, located 0.5 in lateral to the real)	Acupuncture Treatment **11 points:** GV20, GV24, CV6, unilateral, LI 10, HT 7, ST36, SP 6 bilateral Frequency: 2/week during 6 weeks = 12 Treatments Depth of insertion: 0.5 to 1 cm Needle stimulation: The needle was twisted three times to the right Time: 30 min	**Assessment Moments:** - Modified Fatigue Impact Scale at 6 weeks - Sleep, mood, quality of life, and maintenance of benefits at 12 weeks - Parkinson’s Disease Sleep Scale (PDSS) - Epworth Sleepiness Scale (ESS) - Apathy Evaluation Scale (AES). - Quality of life (QOL) - Parkinson’s Disease Questionnaire (PDQ-39)	Acupuncture may improve PD-related fatigue, but real acupuncture offers no greater benefit than sham treatments.	YES—constipation

## Data Availability

Not applicable.

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
