# Peer review of "Effectiveness of Acupuncture in Parkinson’s Disease Symptoms—A Systematic Review"

_healthcare, 2022, doi:10.3390/healthcare10112334_

Round 1

Reviewer 1 Report

This is an interesting review trying to assess the efficacy of acupuncture on Parkinson's disease symptoms.The methods are sound but I think AA might better summarize the results of their revision of the studies they examined.

The discussion is quite balanced but AA should be more coincise and more secure in their conclusions about efficacy of acupuncture.

Author Response

We thank the reviewer for the comments that contributed to improve our manuscript.

According to what was suggested we have now reformulated the Abstract.

Nevertheless, we think that what was found in the review does not let us to be more secure about the efficacy of acupuncture, as it is explained in the discussion section. Actually thats why this systemic review is so important, to understand what we need to do from now on. Of course, our aim is now, continuing to explore the lack in research that we found, to try to create good and sustentable articles that can reach that conclusion, with good methodology.

As we explain now in the abstract, the lack of standardization in the clinical trials reviewed makes difficult to obtain clear/secure conclusions.

Text now included in the manuscript (Abstract):

Background: Parkinson's disease (PD) is the second most common neurodegenerative disease. Several pharmacological and surgical therapies have been developed; however, they are accompanied by several adverse effects. So, many patients have been resorting to complementary medicine, namely acupuncture, in the hope of obtaining symptomatic improvements without having disruptive side effects. Therefore, advances in research in this area are very important. This work presents a systematic review of the effectiveness of acupuncture treatments in relieving PD symptoms; Methods: EMBASE, Medline, Pubmed, Science Direct, The Cochrane Library, Cochrane Central Register of Controlled Trials (Central) and Scielo databases, were systematically searched from January 2011 through July 2021. Randomized controlled trials (RCTs) published in English with all types of acupuncture treatment were included. The selection and analysis of the articles was conducted by two blinding authors through Rayyan application; Results: 720 potentially relevant articles were identified; 52 RCTs met our inclusion criteria. After the exclusion of 35 articles, we found 17 eligible. The included RCTs reported positive effects for acupuncture plus conventional treatment compared with conventional treatment alone in the UPDRS score; Conclusions: Although further studies should be carried out according to rigorous methodological designs, acupuncture treatment seems to have a positive effect on PD symptoms Although all the studies reviewed pointed out a positive effect of acupuncture on improving motor and non-motor symptoms in Parkinson's disease, we found great discrepancies regarding the studies’ design and methodology, making difficult any comparison between them.

The same text was send in word file "Please see the attachment."

Concerning the results, we consider that summarization of more information will hide relevant information that it's relevant for the readers understand and to the respective comparison.

Reviewer 2 Report

Thank you for your great review. I have a little point to comment on. 

- It would be nice if you provided the figure of the acupuncture points with naming. 

- What is/are the most responsive symptoms or last longed to responses after acupuncture? if it is possible, please discuss the mechanism.

- Which stage of PD patients responded well? 

- I agree that we need further well-designed studies for this point. 

Author Response

We thank the reviewer for the comments that contributed to improve our manuscript.

According to what was suggested we have now reformulated our final document.We added at section 3.3.1 the figures of the most common points used, as you recommend, in order to offer a better understanding (Please see the attachment).

We found, like said in discussion, that the patients responded positively to both, motor and non-motor symptoms, however its not possible to compare the response between the most evident symptoms, due the variation in methodology and the poor outcome measures used, that in almost all the studies are general scales and not objective outcome measures. However that question is very pertinent, and we believe that should be studied and explored in future studies.

We suppose that as more advanced the pathology, more deficits patients report, and, we can speculate that better could be their evolution. However, the reviewed studies don't make the intergroups differentiation, in order to make us do that.

Reviewer 3 Report

This is a very interesting study that aimed to investigate the effectiveness of acupuncture in relieving PD symptom. The study is relevant, original and offers a contribution to the knowledge of the area. In that sense, I congratulate the authors. The manuscript, however, needs to be extensively revised and some suggestions and questions are presented here for good application of findings.

1) First impression: Why haven’t you performed a meta-analysis? A meta-analysis would increase the quality of the study and the interest of readers. I strongly suggest the authors to include such analysis. In addition, I highlight (see item 11 of this report) several manuscripts involving this thematic. I am not sure where is the originality of this study in face of other systematic reviews and meta-analysis recently published.

2) Abstract: Avoid repeating terms, such as “several” (rows 15 and 16). Avoid starting sentences with “So”.

3) Introduction: Standardize the use of “Parkinson” (title) or “Parkinson’s” (introduction section).

4) Introduction: Avoid repetition of terms, such as “several” (rows 45 and 47). There are other non-pharmacological treatments (as physical therapy, exercise, occupation therapy, etc) that are interest to be cited in the introduction section.

5) Introduction: Rows 65-68 need a reference.

6) Methods: Please include, in the inclusion criteria, the timeline of year of publication of the manuscript.

7) Methods: I am not sure if the term “Types of participants” is appropriate.

8) Results: Please better detail the points of acupuncture (rows 227 to 238)

9) Results: Is it possible to include a meta-analysis with the efficacy rate (item 3.3.3.) and outcome measure (item 3.4.)?

10) Results: Table 1 – what GC refers?

11) Results: Table 1 – include the values of effect size of the interventions.

12) Results and Discussion: Several articles should be included in the results section and discussed:

12.1.) Effect of Acupuncture on Movement Function in Patients with Parkinson's Disease: Network Meta-Analysis of Randomized Controlled Trials. Healthcare (Basel). 2021 Nov 5;9(11):1502. doi: 10.3390/healthcare9111502.

12.2.) Acupuncture-Related Therapies for Parkinson's Disease: A Meta-Analysis and Qualitative Review. Front Aging Neurosci. 2021;13:676827. doi: 10.3389/fnagi.2021.676827.

12.3.) Electroacupuncture in treatment of Parkinson disease: A protocol for meta-analysis and systematic review. Medicine (Baltimore). 2021 Jan 22;100(3):e23010. doi: 10.1097/MD.0000000000023010.

12.4.) The effectiveness of acupuncture for Parkinson's disease: An overview of systematic reviews. Complement Ther Med. 2020 May;50:102383. doi: 10.1016/j.ctim.2020.102383.

12.5.) Effectiveness and safety of acupuncture combined with Madopar for Parkinson's disease: a systematic review with meta-analysis. Acupunct Med. 2017 Dec;35(6):404-412. doi: 10.1136/acupmed-2016-011342.

12.6.) Effectiveness and safety of acupuncture in the treatment of Parkinson's disease: A systematic review and meta-analysis of randomized controlled trials. Complement Ther Med. 2017 Oct;34:86-103. doi: 10.1016/j.ctim.2017.08.005.

12.7.) Clinical effectiveness of acupuncture on Parkinson disease: A PRISMA-compliant systematic review and meta-analysis. Medicine (Baltimore). 2017 Jan;96(3):e5836. doi: 10.1097/MD.0000000000005836.

13) Discussion: Please improve the discussion topic with the above references.

14) Conclusion and references: ok

Author Response

We thank the reviewer for the comments that helped us to make our manuscript more robust. Thank you very much for your encouraging opinion about our review, considering it as: “relevant, original and offers a contribution to the knowledge of the area”.

With respect to the first point, related to the performance of a meta-analysis, we think that a systematic review of the literature constitutes an adequate methodology for the evaluation of a group of data simultaneously and, as it is well established, it is frequently used to obtain scientific evidence about health interventions. We agree that after a systematic review, a meta-analysis will be an elegant way to present the statistical summary of the data. But the main disadvantage of a meta-analysis is the fact that it cannot compensate for the limitations inherent to the studies in which it is based, since the studies themselves have systematic and aleatory errors that are not corrected by the joint analysis. On the contrary, in that way those errors will be added (Ressing et al, 2009). As we referred to in the discussion, what we found in the review is a great lack of standardization regarding the studies’ design and methodology, making it difficult to compare them. So, we decided to make a systematic review of the literature without a meta-analysis, taking into account that, according to Sampaio et al, 2007, a systematic review, with or without meta-analysis, can be used to present the best evidence for both research and clinic.

Regarding the second, third and fourth point we made changes, as you recommend, marked in the text (attached below - please see the attachment). We standardized Parkinson instead of Parkinson's.

With respect to the fifth point, actually the reference was there, but unfortunately in the wrong place. We already made the alteration. 

Besides, in the methods, we included in inclusion criteria the timeline of year of publication of the manuscript, and changed the terminology too in order to make it more concise. 

Regarding to the eight point, between the rows 231 to 238, and according to what was suggested, we have now reformulated our final document, and added at section 3.3.1 the figures of the most common points used, as you recommend, in order to offer a better understanding (Please see the attachment).

In respect to the results presentation, in Table 1, CG means Control Group and we added the full extension meaning there. Unfortunately the values of the effect size were not presented in most of the articles studied, so we missed that information in order to standardize it. 

Last, but not the least, we included in the results and discussion the articles so well mentioned by you, in order to improve the content. 

One more time, we thank your contributte, and the time and effort spent to help us to enhance this work. 

Round 2

Reviewer 3 Report

I am satisfied with this new version.